# Access to domestic violence advocacy by race, ethnicity and gender: The impact of a digital warm handoff from the emergency department

**Laura Brignone** *, **Anu Manchikanti Gomez**

Sexual Health and Reproductive Equity Program (SHARE), University of California, Berkeley, Berkeley, California, United States of America

* laura.b@berkeley.edu

## Abstract

### Background

Racial/ethnic minority survivors of domestic violence (DV) referred from emergency departments (EDs) face barriers connecting with advocacy services due to systemic inequities. This study assesses the impact of Domestic Violence Report and Referral (DVRR), a digital mandatory reporting compliance system with integrated direct-to-advocacy referrals sent with patient consent, on survivors' likelihood of receiving advocacy by race/ethnicity and gender.

### Methods

We assessed ED charts between 2014–2018 in an urban, Level 1 trauma center for DV-related positive screening, chief concern, or documentation in medical/social work notes. We matched these visits by name to local DV advocacy agency records. Using logistic regression, we assessed survivor odds of reaching advocacy by DVRR receipt, race/ethnicity and gender. We used predicted probabilities to identify between-group differences in advocacy services received.

### Results

Of the 1366 visits, 740 received the DVRR intervention; 323 received advocacy services Without DVRR, compared to white, female survivors (n = 87), male survivors (n = 132) had 0.20 (95% CI: 0.07–0.56) times the odds of reaching advocacy compared to female survivors; Latinx survivors (n = 136) had 2.53 (95% CI: 1.58–4.07) times the odds of reaching advocacy compared to white survivors. With DVRR, the odds and predicted probabilities of reaching advocacy increased significantly for all survivors. White, female survivors (n = 74) who received DVRR experienced 2.60 (95% CI: 1.66–4.07) times the odds of connecting with advocacy. Compared to this reference group, Black survivors (n = 480) who received DVRR experienced 4.66 additional times the odds of reaching advocacy services (95% CI: 3.09–7.04) and male survivors (n = 84) experienced 8.96 additional times the odds (95% CI:

**Data Availability Statement:** All relevant data are within the paper and its Supporting information files.

**Funding:** LB received funding for this research from Gilead Inc. Grant #00495. The funders played no role in the study design, data collection or analysis, decision to publish or preparation of the manuscript. Their website can be accessed here: https://www.gilead.com/purpose/giving/funding-requests.

**Competing interests:** This project received funding from Gilead Inc. Grant #00495. The funder played no role in employment, consultancy, patents, products in development, marketed products, study design, data collection, analysis, or any other project activity. This funding does not alter our adherence to PLOS ONE policies on sharing data and materials. We have no other competing interests to disclose.

2.81–28.56). Overall, we predict 43% (95% CI: 38–53%) of Latinx survivors (n = 177), 36% (95% CI: 31–40%) of Black survivors (n = 480) and 23% (95% CI: 14–32%) of white survivors (n = 83); 37% (95% CI: 33–40%) of women (n = 656) and 29% (95% CI: 18–42%) of men (n = 84) received advocacy services with DVRR.

## Conclusion

Direct-to-advocacy ED referrals facilitated by eHealth technology improve access to advocacy care for all survivors in this sample; marginalized racial and ethnic groups most often victimized by DV appear to have the highest rates of connection to advocacy.

## Introduction

Approximately one in four women has been victimized by physical or sexual violence from an intimate partner during her lifetime, and many more have been harmed by family members, caregivers, or roommates [1]. Factors such as isolation, depression, post-traumatic stress disorder, internalizing responsibility for the perpetrator's abuse, and fear for their safety or legal status have dissuaded domestic violence (DV) survivors from seeking help [2–4]. Nevertheless, nationally representative studies suggest that between 28,000 and 120,000 survivors were seen in emergency departments (EDs) for DV-related chief concerns annually and the authors noted that these were likely vast underestimates due to inconsistent screening and the underutilization of DV-related diagnostic codes [5, 6].

ED providers in multiple studies expressed concern that they were unable to address DV survivors' underlying danger, noting that they lacked training and resources to effectively offer non-medical interventions to DV survivors [7–12]. Standard care for DV includes identification of DV, but rarely includes providing support, facilitating access to support, or following up to see if the patient ultimately received support [13, 14]. When it does, it typically consists of printed educational material or a phone number to a community-based advocacy agency that previous research suggests survivors feel a need to hide from the abuser [15]. In addition, survivors frequently experience post-traumatic stress that inhibits long-term planning [16, 17], and manipulation and coercion from the abuser [3] among other factors that compromise their ability to follow up with these resources. To address these challenges, some hospitals have implemented a strategy known as a warm handoff, in which the ED provider describes ED advocacy agency services personally transfers the survivor's DV care to a DV advocate, typically via a phone call with the patient in the room or an advocate arriving in-person to the patient's bedside to begin offering care [18, 19]. This approach, while resource-intensive, appears to successfully connect survivors to advocacy care [19, 20]. Warm handoff interventions for domestic violence have not been studied by race, ethnicity, gender, and other factors, although survivor experiences with DV victimization, help-seeking and intervention vary among these groups [21–24].

### Inequities by race, ethnicity and gender

In 2017, the Centers for Disease Control and Prevention reported higher victimization rates among individuals who self-identified as Hispanic (all races) (8.6%), Black (9.4%), American Indian/Alaska Native (8.2%) or multiracial (12.5%), compared to non-Hispanic whites (5.7%) in the preceding 12 months [25]. Further, according to the National Health Interview Survey,

Asian, Black, and Hispanic (Latinx)-white patients all experienced significantly less access to healthcare services than non-Hispanic white patients even after adjusting for insurance status [26]. Further, even when survivors of color access healthcare, they appear to experience differential quality of care: a cross-sectional study of 484 medical students found that they systematically discounted the pain, distress and discomfort of non-white survivors and adjusted their treatment recommendations accordingly [27]. Yet a systematic review of 36 studies focused on DV and health outcomes among racial and ethnic minority women found insufficient sample sizes to accurately represent racial or ethnic differences and that these studies often confounded race and ethnicity with environmental and other social determinants of health [28].

Social dynamics surrounding gender further affect survivors' experiences of DV and receipt of medical and advocacy care. A cross-sectional survey of over 10,000 American adults found that 2.5 times more women than men experience DV [25], and 4 times more require medical care for an DV-related injury [29]. Women account for an even larger proportion of DV-coded ED visits: as many as 93–94%, according to nationwide U.S. surveys [5, 30]. Female DV survivors experience a greater likelihood of injury, sexual assault, fear, and depression than male survivors [31]. In addition, consistent barriers to accessing quality healthcare for women carry unique costs for survivors of DV. A landmark review of clinical and experimental research found that healthcare providers tend to systematically disbelieve or downgrade women's self-reports of pain and distress by female patients [32]—a bias providers extend toward elderly and non-white patients as well [33]. Very little of this research has included the experiences of transgender and gender non-binary survivors of domestic violence, despite the additional barriers they face [34].

In contrast, while male survivors of DV have comparatively better access to medical care, they seek and receive DV advocacy care less frequently than female survivors [35]. Two systematic reviews of men's experiences with DV, or help-seeking after DV, suggest this difference derives from men's reluctance to acknowledge abuse victimization, beliefs that DV services are unavailable to male survivors of DV, and fears of professionals not believing them or that they might be accused of perpetrating abuse [36, 37]. In some cases, these fears may be founded. In a qualitative study, both male DV survivors and DV service providers express challenges to men's help-seeking and providers' offering supportive services [38]. In addition, a 2015 literature review of the DV experiences of men suggests many men seek help after experiencing retaliatory violence from a partner against whom they have committed DV [31].

## Domestic Violence Report and Referral (DVRR) intervention

A novel digital intervention, Domestic Violence Report and Referral (DVRR), offers a digital warm handoff for survivors between ED providers and community-based DV advocates via a web-based platform. It is completed by doctors, nurses, social workers, or other care team members in a private setting with no ED visitors present and is available to providers at their discretion as an alternative to faxed, paper-based mandatory reports. DVRR includes body maps to record the nature, images and treatment of injuries; it also includes the 20-question Danger Assessment, a validated questionnaire that predicts a survivor's risk of being killed by their intimate partner [39]. Answers to all Danger Assessment and other DVRR questions are required prior to form submission. These features are guided to bridge any gaps caused by ED providers' lack of DV training [9, 40]. Upon completion, DVRR sends a digital referral to local law enforcement in compliance with California's mandatory reporting requirement, which stipulates that medical professionals who encounter injuries caused by DV must report them to law enforcement [41]. Alongside this mandatory report, DVRR offers an optional automated referral to a local domestic violence advocacy agency. With survivor consent, DVRR

sends their Danger Assessment score and referral information to a local DV advocacy agency, including a safe phone number at which the survivor can be reached by the advocate. DVRR is one of very few DV interventions to include a warm handoff to advocacy or to enable advocates to initiate contact with survivors in the days or weeks after their ED visit [42]. In addition, DVRR is the only mandatory reporting intervention to include a referral or warm handoff to DV advocacy.

A previous analysis of data at three hospitals suggests DV-affected ED patients who received DVRR were over three times as likely to receive subsequent advocacy services [43]. In this paper, we assess the impact of this digital warm handoff referral on survivors' odds and predicted probabilities of receiving advocacy services after an ED visit for DV by race, ethnicity and gender. We compare these findings between groups to determine any differential impact of this intervention on survivors' receipt of advocacy services by race, ethnicity and gender.

## Methods

### Data

This study draws on data collected between February 2014 and April 2018 from a Level 1 trauma center ED and a large DV advocacy center. The Committee for the Protection of Human Subjects at the University of California, Berkeley, as well as the institutional review board of the hospital, approved the study protocol. Trained research assistants at the hospital and trained advocates at the advocacy agency collected data from survivors' electronic health records (EHR) and agency records using a standardized abstraction form. A built-in EHR search program identified all ED visits in which a patient met one of three inclusion criteria: they answered "yes" to the DV screening question "are you being physically hurt or threatened by someone close to you in your living situation?," they received an ICD-9 or ICD-10 (diagnostic) code related to DV, or they stated a chief concern related to DV (e.g., arm broken in fight with boyfriend). These survivors were considered to have "Known DV;" this included violence perpetrated by current or former intimate partners as well as first-degree family members (e.g., siblings, grandparents) and roommates. We included both intimate and non-intimate partner domestic violence as both types are included in California's mandatory reporting law, making both groups equally eligible for DVRR intervention. We documented the number of visits by patients who qualified for study inclusion more than once. These survivors may have required separate intervention for distinct DV episodes, or their visit may have been a follow-up or continuation of an earlier visit; because this distinction could not be determined for repeat survivors, we document the number of visits by patients who qualified for study inclusion more than once. We also noted the number of DV-related visits that included sexual assault, as these individuals received an intervention protocol specific to sexual assault that included referrals to services and advocacy intervention independent of DVRR. Research assistants reviewed all qualifying medical records and abstracted information including the survivor's gender (male/female), race (white, Black), ethnicity (Latinx, non-Latinx), the relationship between perpetrator and victim (e.g., boyfriend, partner, spouse), ED visit date, and whether the DVRR intervention was administered. The researchers provided the advocacy agency with a list of the survivors' full names; the list was grouped by the survivor's race and/or ethnicity, gender and hospital visit date. The agency searched advocacy records using first and last name to determine whether the survivor had received agency services within six months of their hospital visit. To protect the confidentiality of the clients, the agency provided researchers with aggregate client information by race, ethnicity, gender and the date range of their visit. For this reason, age could not be included in the final aggregated dataset.

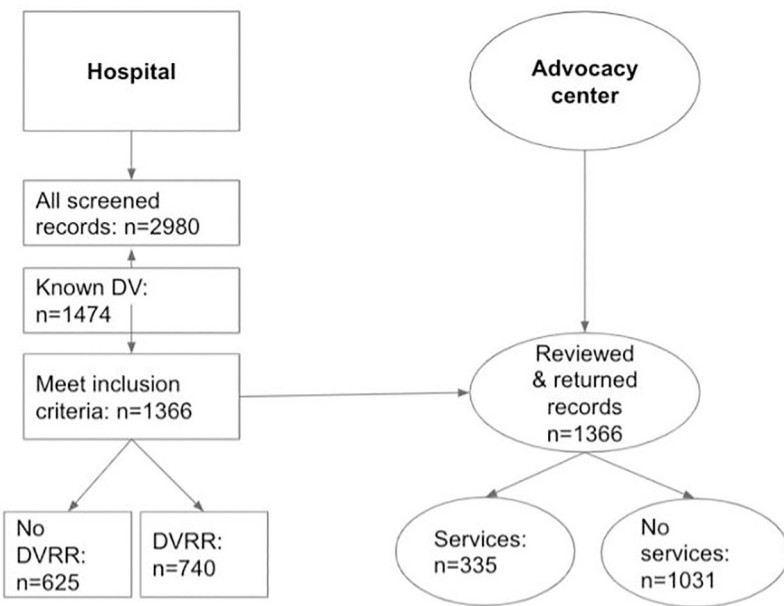

**Fig 1. Selection flow of medical and advocacy records included for analysis.**

We excluded the records of survivors who did not identify in the included racial (white/ Black), ethnic (Latinx/non-Latinx) and gender (male/female) categories (n = 111) due to the small sample sizes of these groups. We documented DVRR administration based on a completed report in the online system; we also considered that survivors with a partial, incomplete DVRR record had not received DVRR. We detected no missing or incomplete data within the hospital dataset; due to its aggregate nature the quality of the advocacy data is impossible to verify. Our final sample included 1366 survivor ED visits. Fig 1 outlines the chart selection process.

## Analysis

We used chi-square tests to detect significant differences in receipt of DVRR by race, ethnicity, gender, sexual assault victimization and repeat visit status. We used logistic regression to assess the association between receiving DVRR and the odds of receiving advocacy services. The logistic regression model included indicator variables for survivor race, ethnicity (Black, Latinx, white) and gender (female, male). We also included two interaction terms, one between DVRR administration and gender and the other between DVRR administration and racial/ ethnic group. Stata's *lincom* command was used to estimate the linear combination of coefficients. For each category of race, ethnicity and gender, we assessed the impact of DVRR on the survivor's eventual connection to advocacy services using predicted probability with 95% confidence intervals. This offers a straightforward way to compare the differences between group likelihood of receiving advocacy with vs without DVRR. To accomplish this, we measured differences in predicted probability between groups, and tested those differences for significance using chi square tests. We conducted statistical analyses using Stata 14.2.

## Results

### Characteristics of the study population

Between February 2014 and April 2018, medical records for 1366 survivor visits to the ED documented a diagnosis, chief concern or medical/social work note that indicated a DV episode

**Table 1. Descriptive characteristics of study ED visits.**

| | *(b) Total (n = 1366)* | | (a) DVRR administered (n = 740) | | (a) DVRR not administered (n = 626) | |
|---|---|---|---|---|---|---|
| | **%** | **N** | **%** | **N** | **%** | **N** |
| **Gender** | | | | | | |
| Female** | *84.2%* | 1150 | 88.6% | 656 | 78.9% | 494 |
| Male | *15.8%* | 216 | 11.4% | 84 | 21.1% | 132 |
| **Race and Ethnicity** | | | | | | |
| Black | *63.5%* | 868 | 64.9% | 480 | 62.0% | 388 |
| White | *13.5%* | 185 | 11.2% | 83 | 16.3% | 102 |
| Latinx | *22.9%* | 313 | 23.9% | 177 | 21.7% | 136 |
| **Sexual Assault** | | | | | | |
| Yes** | *14.9%* | 203 | 12.2% | 90 | 18.1% | 113 |
| No | *85.1%* | 1162 | 87.8% | 650 | 81.8% | 512 |
| **Multiple DV visits** | | | | | | |
| Yes | *25.7%* | 351 | 24.5% | 181 | 27.2% | 170 |
| No | *74.2%* | 1014 | 75.5% | 559 | 72.7% | 455 |

*$p < 0.05$;

**$p < 0.01$.

precipitated the ED visit. The majority of survivors were female (84.2%), identified as Black or African American (63.5%), did not experience sexual assault during the DV episode (85.1%), and had no other DV-related visits to the trauma center during the study period (74.2%). Providers administered DVRR in 54.2%, or 740, of the study visits (Table 1). Chi-square tests comparing the characteristics of survivors who received DVRR and those who did not revealed that the groups were similar in terms of race, ethnicity and repeat visit status. However, significantly more female than male survivors received the intervention. Providers were also significantly less likely to administer the intervention to survivors who had also experienced sexual assault during the DV episode.

## DVRR and receiving advocacy services by race, ethnicity, gender

DVRR administration was associated with a significant increase in all survivors' odds of receiving advocacy services (Table 2). At baseline, white, female survivors' odds of reaching advocacy services were 0.12 (95% CI: 0.07–0.20); Black survivors experienced similar odds. Latinx survivors experienced significantly higher odds of connecting with advocacy services than Black and white survivors (OR: 2.53, 95% CI: 1.58–4.07). Black survivors who did not receive the intervention had approximately the same odds of receiving services as white survivors. White women who received the intervention experienced an estimated 2.60-fold (95% CI: 1.66–4.07) increase in their odds of connecting with advocacy services. No significant or meaningful difference in odds was detected for Latinx survivors who received the intervention compared to white female survivors who received it; to preserve power in the model, no separate odds ratio was calculated for this group. A significant interaction (DVRR x Black) indicated that DVRR was associated with a greater change for Black survivors than for white women or Latinx survivors; Black survivors who received the intervention experienced an *additional* 4.67-fold (95% CI: 3.09–7.04) increase in the odds of connecting to advocacy services. Male DV survivors who did not receive the intervention had 0.20 times (95% CI: 0.07–0.55) the odds of connecting with advocacy services compared to female survivors. However, a significant interaction

**Table 2. Odds ratios of connection to advocacy by patient receipt of DVRR, race/ethnicity and gender.**

|  | *No DVRR* | | *DVRR* | | *Interaction terms* |
|---|---|---|---|---|---|
|  | *n* | *OR* | *n* | *OR* | *OR for DVRR = 1 vs DVRR = 0* |
|  | *advocacy / no advocacy* | *(95% CI)* | *advocacy / no advocacy* | *(95% CI)* |  |
| *White, Female* | 9/78 | Reference | 19/55 | 2.60** (1.66–4.07) | N/A |
| *Black* | 34/354 | 1.00 (0.54–1.82) | 168/312 | 1.80* (0.99–3.26) | 4.67** (3.09–7.04) |
| *Latinx* | 28/108 | 2.53** (1.58–4.07) | 76/101 | Included in reference | N/A |
| *Male* | 4/132 | 0.20** (0.07–0.56) | 24/60 | 3.45* (1.09–10.87) | 8.96** (2.81–28.56) |

*p<0.05;

**p<0.01.

(DVRR x male) indicated that DVRR was associated with a greater change for male survivors than for female survivors. Male survivors who received DVRR experienced an *additional* 8.96-fold (95% CI: 2.81–28.56) increase in the odds of connecting with advocacy services.

Next, we assessed the predicted probability of Black, Latinx, white, female and male survivors receiving advocacy services with and without the intervention. Without the intervention, 13% of Black, 20% of Latinx, and 10% of white survivors received follow-up services (Fig 2). When DVRR was administered, more than twice as many Black, Latinx and white survivors were predicted to reach advocacy, and these increases were significant within each group. This resulted in 29% of Black survivors, 43% of Latinx survivors and 23% of white survivors reaching advocacy. With the intervention, Latinx survivors were predicted to have a significantly higher probability of reaching advocacy services than white survivors (PP: 1.78; 95% CI: 1.17–2.71; p<0.01, results not shown).

Female survivors were significantly more likely than male survivors to subsequently connect with advocacy services (Fig 3). DVRR use appears to narrow that gap within this sample. Without DVRR, 16% of women and 4% of men subsequently received advocacy services, a significant difference (Likelihood difference: -9.46%; 95% CI: -13.08% to -5.82%; p<0.01). With DVRR administration, more than twice as many women and six times as many men received advocacy services, resulting in 33% of women survivors and 25% of men survivors receiving advocacy services (Likelihood difference: -9.16%; 95% CI: -18.89 to 0.57%; p = 0.06).

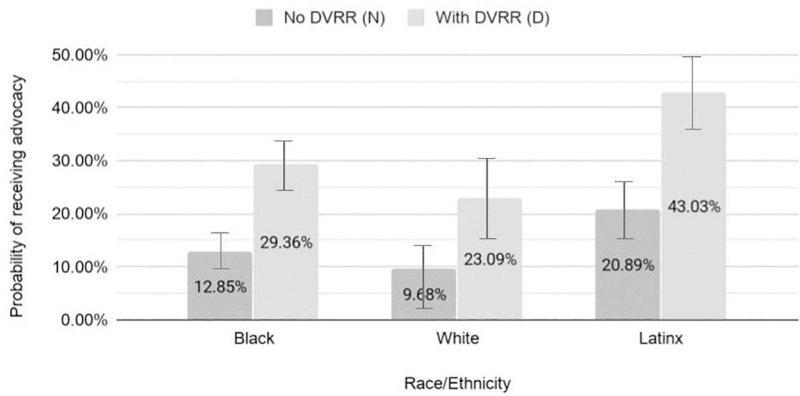

**Fig 2. Predicted probability (95% CI) of follow-up advocacy services by DVRR status and race and ethnicity.**

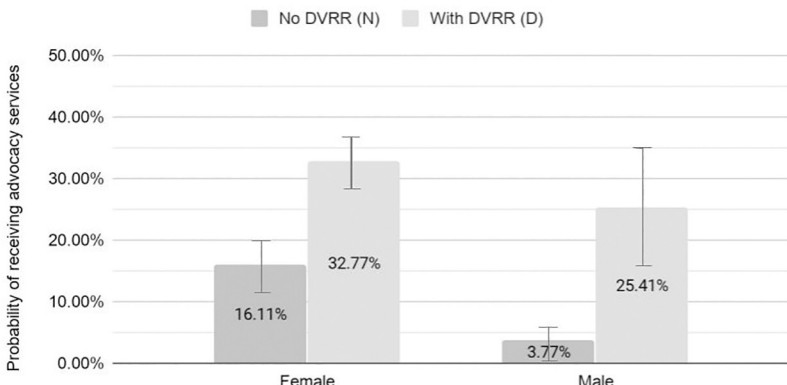

**Fig 3. Predicted probability (95% CI) of follow-up advocacy services by DVRR status and sex.** (see S1 Appendix for detailed results).

## Discussion

In this study, receipt of the DVRR digital warm handoff intervention increased patient odds and rate of connection to DV advocacy services for all survivors. It facilitated the highest rates of connection to advocacy among the groups most likely to be affected by DV: an estimated 43% of Latinx survivors and 29% of Black survivors reached advocacy when given DVRR [25]. Increased odds of reaching advocacy services were significantly greater for Black than white survivors due to the interaction between DVRR and Black identity.

DVRR was not developed to address inequities in DV care. However, many of the mechanisms through which this digital warm handoff intervention operates appear to correspond to strategies identified as culturally appropriate among Black and Latinx survivors. For Latinx survivors, several studies highlight the importance of interpersonal solidarity, cultural connections and family support in fostering safety and resilience [44–46]. Related themes of cultural solidarity and resisting victimization narratives are highlighted in research with Black survivors [22, 24, 47]. Warm handoff interventions such as DVRR facilitate personal connections between the advocate and survivor. In addition, warm handoffs may protect survivors against needing to embrace a stance of victimization and weakness that may conflict with cultural norms in order to reach advocacy services [22, 24, 48]. These factors may make digital warm handoff interventions uniquely applicable to Black and Latinx survivors and may at least partially explain the large effect size among these groups in this study.

Another mechanism for these changes may include a reduction in provider bias among delivery of the intervention to survivors. Historical scholarship and experimental research that suggest providers are more likely to stereotype and discount experiences of pain among racial and ethnic minorities (45,60). The structured questions in this intervention may bypass these providers' unconscious biases and increase non-white survivors' likelihood of reaching follow-up services. By including the Danger Assessment as the basis for its law enforcement report and advocacy referral, this digital warm handoff standardizes information-gathering and automates the content, delivery and destination of referrals. Currently, the 20-question Danger Assessment form, validated for women in heterosexual abusive relationships, is used for all DVRR recipients; future developments of DVRR include the brief, 5-question Danger Assessment tool that is validated across genders and racial and ethnic groups. This is an important step toward improving the strength of the Danger Assessment component of DVRR in reducing provider bias and providing culturally appropriate care. Any effect of DVRR in reducing

provider bias would only apply to providers who chose to use DVRR. A shortened (5-question) Danger Assessment may shorten the provider time required to complete DVRR, thus increasing the likelihood of its use from 54% to an even higher value.

DVRR was also associated with a significant increase in receipt of advocacy services among both men and women. An estimated 33% of women and 25% of men reached advocacy services when given DVRR. This suggests that the intervention was associated with a reduction in the barriers that prevented both men and women survivors from reaching care. For women, the standardized Danger Assessment questionnaire may limit provider bias that discounts the severity of women's experiences [32]. While the Danger Assessment questionnaire has not been validated in men, the structured, direct-to-advocacy referral may circumvent a self-reported reluctance to report or seek help and it may bypass potential fears that they may be ineligible for services [48, 49]. Providers administered DVRR at their discretion, so providers who chose to administer DVRR may also have been more likely to proactively connect survivors to services.

## Strengths and limitations of the study

This study is the first to examine the impact of a digital warm handoff to DV advocacy on survivor outcomes by race and ethnicity and by gender. It contributes to the small bodies of literature exploring eHealth interventions for DV in the ED and warm-handoffs for DV survivors [50]. Additionally, it is one of the few to explore how digital interventions facilitate a warm handoff for DV survivors [42, 43]. Our findings highlight digital warm handoffs' potential to provide additional benefit to the survivors most negatively impacted by DV. This study is further unique within DV research as its methods confidentially link hospital and advocacy records of DV survivors.

Only one hospital was included in this study, so the primary limitation of this study stems from its lack of generalizability. In addition, the matching technique between hospitals and advocacy records relies on first and last names, which could give groups with a higher frequency of identical given names (e.g., Latinx, male) a higher risk of false positive matches within this sample. Because records were matched based on survivor first and last names, spelling variability between hospital and advocacy records may also compromise the validity of the matched records.

To protect client privacy at the advocacy agency, covariate data collection was restricted *a priori* to variable combinations with cell sizes of 2 or greater. As a result, researchers were limited to six major covariate indicators (DVRR administration, male or female gender, Black, Latinx or non-Latinx white race/ethnicity). This required excluding certain populations with small sample sizes (e.g., Asian, Pacific Islander, Indigenous), and not specifying other relevant characteristics either due to a small sample size (e.g., LGBTQ+ identity) or non-categorical data structure (e.g., age). State-level studies, nationally representative research, and systematic reviews of smaller studies suggest many of these groups experience heightened barriers to DV response or services. These include survivors who are Asian, Pacific Islander, and Indigenous [28, 50] queer, transgender, and non-binary [51, 52] elderly [53] or who individuals live in neighborhoods with violence, limited access to services and other forms of community trauma [29, 54]. Future research may examine the efficacy of digital warm handoff interventions in connecting these DV survivors to advocacy services. In addition, the hospital in this study saw a majority of Black IPV patients (63.5%); this proportion is likely higher than in many other hospitals and may be associated with a higher level of culturally appropriate healthcare. The impact of DVRR for racial and ethnic minority patients in hospitals with a less heterogenous patient population may differ from the findings of those in this study due to potential

differences in care. Future research should investigate the role of DVRR in reducing inequities among a broader sample of hospitals to determine the generalizability of the present study's findings. As DVRR administration is contingent on provider discretion, future research should determine if providers differentially administer DV care, including DVRR, and any role this may play in survivor outcomes.

Finally, though rates of advocacy contact are significantly higher with a digital warm hand-off, they are still quite low. Despite a two-fold or higher effect size for each group, fewer than half of all DV survivors in this sample ultimately reached contact with advocacy. Reasons for this may range from non-working phone numbers to unanswered phone calls. These missed connections may have resulted from survivors changing their mind about receiving advocacy care, the abuser monitoring the survivor's phone, or inadvertent typos or missed calls. Future research should investigate the outcomes of survivors who do not connect with advocacy, including their preferences, needs, and interventions that may help them achieve safety, as well as the impact of connection to advocacy services. Any differences between survivors harmed by an intimate partner versus a family member or roommate should also be explored.

## Conclusion

The present study found that a digital warm handoff referral to DV advocacy improved access to care for women and men and among Black, Latinx, and white survivors, with a significant additional increase in the odds of advocacy connection among Black and male survivors. These findings suggest that a digital warm handoff provides meaningful assistance to all DV survivors; they further suggest that such an intervention can be particularly meaningful for members of groups at the greatest risk of DV and inequities in care. Given the disproportionate burden of DV on these vulnerable groups and the additional barriers they face in accessing adequate DV and healthcare, DVRR represents a meaningful step toward adequate support for these vulnerable survivors as they seek DV care in an ED setting.

## Supporting information

**S1 Appendix. Standardized data abstraction form.**
(PDF)

**S2 Appendix. Related manuscript—Main effects.** Domestic Violence Report and Referral: A Multiple Baseline Study of an eHealth Warm Handoff for Emergency Department Patients affected by Domestic Violence.
(PDF)

**S1 Dataset. Minimum underlying dataset.**
(XLSX)

## Acknowledgments

The authors would like to thank Hillary Larkin and the participating hospital for granting access to DVRR and hospital data, the participating advocacy agency for providing data and assisting in abstraction, and the numerous research assistants who helped abstract hospital data. The authors would also like to thank Jeffrey Edleson and Emily Ozer for their reviews of earlier versions of this manuscript.

## Author Contributions

**Conceptualization:** Laura Brignone.

**Data curation:** Laura Brignone.

**Formal analysis:** Laura Brignone.

**Funding acquisition:** Laura Brignone.

**Investigation:** Laura Brignone.

**Methodology:** Laura Brignone.

**Supervision:** Laura Brignone, Anu Manchikanti Gomez.

**Writing – original draft:** Laura Brignone.

**Writing – review & editing:** Laura Brignone, Anu Manchikanti Gomez.

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
