## [Decision Letter · Decision Letter 0]

27 May 2021

PONE-D-21-05014

Access to domestic violence advocacy by race, ethnicity and gender:

The impact of a digital warm handoff from the emergency department

PLOS ONE

Dear Dr. Brignone,

Thank you for submitting your manuscript to PLOS ONE. After careful consideration, we feel that it has merit but does not fully meet PLOS ONE’s publication criteria as it currently stands. Therefore, we invite you to submit a revised version of the manuscript that addresses the points raised during the review process.

Thank you for submitting this manuscript on a very important topic related to IPV and racial differences in receiving and accessing advocacy services. This manuscript fills an important gap, but needs some additional editing and clarity before publication. 

We look forward to receiving your revised manuscript.

Kind regards,

Michelle L. Munro-Kramer, PhD, CNM, FNP-BC

Academic Editor

PLOS ONE

Additional Editor Comments:

Both of the reviewers provided detailed comments and suggestions that should be attended to prior to resubmission. I would specifically encourage the authors to pay attention to:

1) Streamlining the background section to focus specifically on the topics related to this study. Please see Reviewer #1's suggestion for how to do this.

2) Reviewer #1 & Reviewer #2's questions related to clarification in the methods section (e.g., which Danger Assessment was used - 5 versus 20 items; what EHR search program; how referral data was linked; how are repeat visits handled; measurement/definitions used for race and gender).

3) Discussion - As noted by Reviewer #2 - the study included a high proportion of Black patients. How did this impact the results? Please provide more details/discussion around the fact that about half of the sample did not receive the intervention and of those who did only 23-43% accessed services.

Journal Requirements:

"LB received funding for this research from Gilead Inc. Grant #00495. The funders played no role in the study design, data collection or analysis, decision to publish or preparation of the manuscript. Their website can be accessed here: " ext-link-type="uri" xlink:type="simple">https://www.gilead.com/purpose/giving/funding-requests"

We note that you received funding from a commercial source: Gilead Inc.

Reviewers' comments:

Reviewer's Responses to Questions

**Comments to the Author**

1. Is the manuscript technically sound, and do the data support the conclusions?

Reviewer #1: Partly

Reviewer #2: Yes

2. Has the statistical analysis been performed appropriately and rigorously? 

Reviewer #1: Yes

Reviewer #2: Yes

3. Have the authors made all data underlying the findings in their manuscript fully available?

Reviewer #1: Yes

Reviewer #2: Yes

4. Is the manuscript presented in an intelligible fashion and written in standard English?

Reviewer #1: No

Reviewer #2: Yes

5. Review Comments to the Author

Reviewer #1: This is an important study of ED patients with DV, and their receipt of a digital warm handoff intervention called DVRR, and their receipt of advocacy services. I have several questions and comments to improve clarity.

Abstract, Methods:

1. Can you describe briefly how 1366 ED visits are related to 323 DV advocacy visits? Meaning are the DV advocacy visits from DVRR performed in ED?

2. Is the intervention DVRR? Meaning odds and predicted probability are comparing ED visits for DV where ED patients received DVRR or not? Did all ED patients with DV receive DVRR?

3. Can you explain more what predicted probabilities adds to odds ratios? Is it to show disparities by race/ethnicity of receiving DVRR?

Abstract, Results:

1. Latinx odds reference group is Black and white combined? In a logistic regression? Did you use a multinomial logistic regression with Latinx compared to Black compared to white

2. Is this odds of 2.53 for Latinx survivors who did not receive DVRR?

3. When you write compared to white, female survivors, is Black survivors’ 4.66 odds of reaching advocacy in those without DVRR? Is this a logistic regression (binary outcome) or multinomial outcome?

4. Male survivors experienced 12.8 times the odds compare to who? Female survivors with DVRR?

Manuscript text:

Introduction:

1. The section on race and ethnicity first sentences describes DV race/ethnicity differences. I would keep that. The 5 paragraphs describe differences in use of or service-provider creation of DV advocacy services and shelter. However, the length of text distracts from the focus on ED patients, DVRR those patients do or don’t receive, and eventual services those patients do or don’t use. I would remove the remaining material except for the last paragraph on “examining ED services”.

2. I would keep the section on gender as is, as it’s concise at 2 paragraphs.

3. DVRR section: can you describe more about DVRR? This in many ways is the crucial part of your introduction. I don’t know how to find reference 51, which is listed as a poster. Is it https://urapprojects.berkeley.edu/projects/detail.php?id_list=Soc0795? That describes data to collect, not data about DVRR. See my next set of questions below.

Methods

Intervention

1. I would move this paragraph to introduction. This material says DVRR is used at the discretion of the ED provider. Can you tell us more about this discretion? This gets at provider bias, and in many ways may determine if DVRR is useful or not.

2. Is the Danger Assessment the full 20 questions? Or the shorter versions like at https://pubmed.ncbi.nlm.nih.gov/28921610/? A short vs long questionnaire (5 questions vs. 20 questions) may mean providers use or don’t use DVRR, as an extra 15 questions will increase provider time in working with patients.

3. Describe more about CA mandatory reporting law. It is unlike other state mandatory reporting laws (https://www.futureswithoutviolence.org/userfiles/file/HealthCare/Compendium%20Final.pdf).

4. Do you know how many ED patients with DV receive DVRR, as opposed to provider who calls police directly to make report (for example in other states providers must make a call)?

5. Who (nurse, social worker, doctor) completes DVRR during a clinical encounter? Does that influence if DVRR is used?

6. Do patients not receive other DV assistance during DV encounter (safety planning for positive danger assessment screen) if DVRR is completed (where an advocate later may or may not call patient)?

7. Are ED’s completing DVRR without other ED visitors/potential perpetrators present?

8. Can survivors voluntarily decide not to receive DVRR? I know you may not have answers to many of my questions on DVRR, but the answers all influence how we interpret your results. DVRR is a novel process that I think is great, but it deserves explaining so that providers in other states can think about if a similar program is worth developing.

Data

1. Can we see a copy of the standardized abstraction form? Maybe put it as an appendix?

2. Tell me more about the EHR search program that identified all ED visits in which a patient screened positive for DV. What screening tool was used? Was the answer to only 1 DV question (out of however many DV questions were asked) considered a positive screen?

3. How do you define chief complaint (I recommend calling it a chief concern; providers may feel that patients are complaining, but patients express important concerns about their health) related to DV? Does the patient have to say “I am a victim of DV” to be considered having a chief concern related to DV?

4. Do ED patients with repeated visits for DV receive multiple DVRR?

5. Were visits by survivors of sexual assault by a partner considered to have experienced sexual assault or DV?

6. How is gender measured? Does it include non-binary identifying patients?

7. How is race defined? By patient or provider? What about a patient who has more than 1 race?

8. When you describe “the relationship between perpetrator and victim”, aren’t all DV cases relationships between partners? Is this referring to sexual assault cases?

9. Do you have a measure of how well DVRR was administered? Like full 20 question Danger Assessment, or 5-item Danger Assessment screen described above? Was there safety planning conducted in the ED for positive Danger Assessment screens?

10. Were there duplicates for survivor full names and hospital visit dates at DV agency?

Figure 1: How do you define known DV? What are your inclusion criteria? Do we know why some patients did not receive DVRR? Shouldn’t they all have?

Data analysis

1. Does DVRR administration mean that DVRR was documented to have occurred during the ED visit? Or that a DVRR report existed in the online system? Did patients receive multiple DVRR over multiple visits?

2. Describe more about what predicted probably adds to the logistic regression models

Characteristics of the study population

1. 74.2% had no other visits to the trauma center during the study period. Does that mean no other ED visits? Or no other visits for DV?

2. You write that survivors who experienced sexual assault were less likely to have administered DVRR. Do those providers receive a different intervention?

3. I’d recommend moving this paragraph to results.

Table 1: what are the **denoting? A level of p value? Or just a statistically significant difference in % values in that characteristic?

Results

1. Can you explain how the interaction terms increased the odds of connecting to advocacy services?

2. You write odds of accessing advocacy services, and odds of connecting to advocacy services. Are those the same things?

3. Explain what the predicted probability adds to the odds ratio analyses already presented. Isn’t it showing similar results to DVRR associated with increased odds of receiving advocacy services?

Table 2: is this a multinomial logistic regression? Of a series of logistic regressions shown in different columns? What is the reference group for Baseline odds, No DVRR white female? Why in the DVRR female white and latinx are white and latinx grouped together?

Discussion:

1. 2nd paragraph: You write that DVRR is an automatic referral. Given that only 54% of patients received it (from Methods/Characteristics of the study population), and your later description of providers choosing to administer DVRR may be more likely to connect survivors to services, I think you can describe this as a digital referral with lethality assessment.

2. Is there literature on Danger Assessment being validated in men who experienced violence?

Strengths and limitations

1. You write that “it confidentially links the hospital and advocacy records of DV survivors.” Do you mean DVRR, or this research study on DVRR?

2. Do you have data on how many identical given names (duplicates) occurred in your data? That would allow you to determine if duplicates affected intervention and non-intervention groups equally. I otherwise do not think you can conclude this.

Reviewer #2: Access to domestic violence advocacy by race, ethnicity and gender:

The impact of a digital warm handoff from the emergency department

Accept minor revisions

This study fills an important gap in our evidence base regarding intervention for IPV, and its focus on racial differences is particularly timely. It is also very nice to see an intervention that differentially and positively impacts Latinx and Black women. The intervention itself is well-conceived to address challenges in IPV referral in the ED, and in particular that the hand-off is to an advocacy organization rather than only to law enforcement (as required by CA law).

Specific comments:

1. The introduction is far too lengthy and reads more like an essay. Only the salient points for this analysis, summarized, are needed.

2. The authors state in their methods that they detected no missing or incomplete data, but with only aggregate data from the advocacy organizations, the quality of the data is impossible to verify. This should be noted.

3. The process for linking referrals to follow-up visits is not described. Linkages based on name, because of spelling variability, are particularly problematic and generally have some validity challenges.

4. Given strong evidence that BIPOC patients report challenges in their interactions with healthcare facilities, the generalizability of this study needs to be addressed. For example, many EDs might not have a majority of Black IPV patients (in this sample, 63.5%), and perhaps the staff at this hospital are particularly adept in culturally appropriate and ethnicity-centered healthcare. How would this potentially impact EDs with different patient profiles?

5. Although these findings are very promising, nearly half of the IPV victims in the ED did not receive the intervention, and even with the improved access to advocacy services with the intervention, only between 23% and 43% of those referred accessed services. The discussion should address these issues by providing some insight into why these proportions are low and how they might be improved.

6. PLOS authors have the option to publish the peer review history of their article (what does this mean?). If published, this will include your full peer review and any attached files.

Reviewer #1: No

Reviewer #2: No

---

## [Author Response · Author response to Decision Letter 0]

16 Jun 2021

Both of the reviewers provided detailed comments and suggestions that should be attended to prior to resubmission. I would specifically encourage the authors to pay attention to:

1) Streamlining the background section to focus specifically on the topics related to this study. Please see Reviewer #1's suggestion for how to do this.

The background section has been streamlined and substantially shortened as requested.

2) Reviewer #1 Reviewer #2's questions related to clarification in the methods section (e.g., which Danger Assessment was used - 5 versus 20 items; what EHR search program; how referral data was linked; how are repeat visits handled; measurement/definitions used for race and gender).

These clarifications have been provided and are described in detail in the reviewer responses.

3) Discussion - As noted by Reviewer #2 - the study included a high proportion of Black patients. How did this impact the results? Please provide more details/discussion around the fact that about half of the sample did not receive the intervention and of those who did only 23-43% accessed services.

The discussion section has been expanded to include these details.

 Journal Requirements:

https://journals.plos.org/plosone/s/file?id=wjVg/PLOSOne_formatting_sample_main_body.pdf and https://journals.plos.org/plosone/s/file?id=ba62/PLOSOne_formatting_sample_ title_authors_affiliations.pdf

PLOS guidelines have been reviewed and submission materials have been updated accordingly

 2. Thank you for stating the following in the Financial Disclosure section: "LB received funding for this research from Gilead Inc. Grant #00495. The funders played no role in the study design, data collection or analysis, decision to publish or preparation of the manuscript. Their website can be accessed here: https://www.gilead.com/purpose/ giving/funding-requests." We note that you received funding from a commercial source: Gilead Inc. Please provide an amended Competing Interests Statement that explicitly states this commercial funder, along with any other relevant declarations relating to employment, consultancy, patents, products in development, marketed products, etc. Please include your amended Competing Interests Statement within your cover letter. We will change the online submission form on your behalf.

This statement has been amended as requested and declared in the cover letter.

This statement has been declared in the cover letter.

The authors have declared in the cover letter that we have no other competing interests to disclose.

3. In your Data Availability statement, you have not specified where the minimal data set underlying the results described in your manuscript can be found. PLOS defines a study's minimal data set as the underlying data used to reach the conclusions drawn in the manuscript and any additional data required to replicate the reported study findings in their entirety. All PLOS journals require that the minimal data set be made fully available. For more information about our data policy, please see http://journals.plos.org/plosone/s/data-availability. Upon re-submitting your revised manuscript, please upload your study’s minimal underlying data set as either Supporting Information files or to a stable, public repository and include the relevant URLs, DOIs, or accession numbers within your revised cover letter. For a list of acceptable repositories, please see http://journals.plos.org/plosone/s/data-availability#loc-recommended-repositories. Any potentially identifying patient information must be fully anonymized. We will update your Data Availability statement to reflect the information in your cover letter. Important: If there are ethical or legal restrictions to sharing your data publicly, please explain these restrictions in detail. Please see our guidelines for more information on what we consider unacceptable restrictions to publicly sharing data: http://journals.plos.org/plosone/s/data-availability#loc-unacceptable-data-access-restrictions. Note that it is not acceptable for the authors to be the sole named individuals responsible for ensuring data access.

The minimum underlying dataset is attached as a Supporting Information file. This has been noted in the cover letter.

This information has been updated as requested.

Reviewer #1: This is an important study of ED patients with DV, and their receipt of a digital warm handoff intervention called DVRR, and their receipt of advocacy services. I have several questions and comments to improve clarity.

Abstract, Methods:

1. Can you describe briefly how 1366 ED visits are related to 323 DV advocacy visits? Meaning are the DV advocacy visits from DVRR performed in ED?

The abstract text has been updated to read: “323 subsequent visits to a local DV advocacy agency.” The Methods now clarify that DV advocates respond to the referral within days or weeks after their ED visit. 

2. Is the intervention DVRR? Meaning odds and predicted probability are comparing ED visits for DV where ED patients received DVRR or not? Did all ED patients with DV receive DVRR?

The intervention is DVRR; about half of ED patients with DV received DVRR. This has been clarified in the abstract.

3. Can you explain more what predicted probabilities adds to odds ratios? Is it to show disparities by race/ethnicity of receiving DVRR?

Yes; this has been clarified in the abstract.

Abstract, Results:

1. Latinx odds reference group is Black and white combined? In a logistic regression? Did you use a multinomial logistic regression with Latinx compared to Black compared to white?

This has been clarified in the abstract; white female survivors constituted the odds reference group in this logistic regression. Without DVRR, Black and white survivors reached advocacy at rates that were not significantly different. 

2. Is this odds of 2.53 for Latinx survivors who did not receive DVRR?

Yes. This has been clarified in the abstract.

3. When you write compared to white, female survivors, is Black survivors’ 4.66 odds of reaching advocacy in those without DVRR? Is this a logistic regression (binary outcome) or multinomial outcome?

Logistic regression; this has been clarified in the abstract.

4. Male survivors experienced 12.8 times the odds compare to who? Female survivors with DVRR?

Yes. This has been clarified in the abstract.

Manuscript text:

Introduction:

1. The section on race and ethnicity first sentences describes DV race/ethnicity differences. I would keep that. The 5 paragraphs describe differences in use of or service-provider creation of DV advocacy services and shelter. However, the length of text distracts from the focus on ED patients, DVRR those patients do or don’t receive, and eventual services those patients do or don’t use. I would remove the remaining material except for the last paragraph on “examining ED services”.

The section has been edited per this suggestion and now consists of two paragraphs, comparable in length to the section on gender. 

2. I would keep the section on gender as is, as it’s concise at 2 paragraphs.

Done, thanks.

3. DVRR section: can you describe more about DVRR? This in many ways is the crucial part of your introduction. I don’t know how to find reference 51, which is listed as a poster. Is it https://urapprojects.berkeley.edu/projects/detail.php?id_list=Soc0795? That describes data to collect, not data about DVRR. See my next set of questions below.

This paragraph has been combined with the first paragraph of the Methods section and its detail has been expanded considerably.

Methods

Intervention

1. I would move this paragraph to introduction. This material says DVRR is used at the discretion of the ED provider. Can you tell us more about this discretion? This gets at provider bias, and in many ways may determine if DVRR is useful or not.

This paragraph is now in the introduction. Details about provider discretion are also included in this paragraph. 

2. Is the Danger Assessment the full 20 questions? Or the shorter versions like at https://pubmed.ncbi.nlm.nih.gov/28921610/? A short vs long questionnaire (5 questions vs. 20 questions) may mean providers use or don’t use DVRR, as an extra 15 questions will increase provider time in working with patients.

Right now, the program uses the full 20 questions (added to the Methods). In addition, the discussion now notes this complication and states that future DVRR development is replacing this with the 5-question version for a more efficient, streamlined approach.

3. Describe more about CA mandatory reporting law. It is unlike other state mandatory reporting laws (https://www.futureswithoutviolence.org/userfiles/file/HealthCare/Compendium%20Final.pdf).

Added: Upon completion, DVRR automatically sends a digital referral to local law enforcement in compliance with California’s mandatory reporting requirement, which stipulates that medical professionals who encounter injuries caused by DV must report them to law enforcement.

4. Do you know how many ED patients with DV receive DVRR, as opposed to provider who calls police directly to make report (for example in other states providers must make a call)?

Because it is hospital policy (and a legal requirement) that all DV cases seen in the ED be reported to police, I assume that all survivors who did not receive DVRR did have a phone call made to police on their behalf (and I’ve added that assumption to the Methods). However, we did not measure compliance with the mandate.

5. Who (nurse, social worker, doctor) completes DVRR during a clinical encounter? Does that influence if DVRR is used?

Any member of the care team can complete DVRR (added to methods). Right now we don’t know if that influences whether or not DVRR is used – this is noted in the discussion and a question for the next manuscript.

6. Do patients not receive other DV assistance during DV encounter (safety planning for positive danger assessment screen) if DVRR is completed (where an advocate later may or may not call patient)?

This was not measured; existing hospital protocols don’t suggest that treatment of DVRR recipients would differ from non-DVRR patients. 

7. Are ED’s completing DVRR without other ED visitors/potential perpetrators present?

Added: DVRR is completed without ED visitors for the survivor’s privacy.

8. Can survivors voluntarily decide not to receive DVRR? I know you may not have answers to many of my questions on DVRR, but the answers all influence how we interpret your results. DVRR is a novel process that I think is great, but it deserves explaining so that providers in other states can think about if a similar program is worth developing.

Survivors can decide to not answer questions and can decline to have the report sent to advocacy; they can’t prevent the mandatory report from being sent. This has been clarified in the text.

Data

1. Can we see a copy of the standardized abstraction form? Maybe put it as an appendix?

Attached.

2. Tell me more about the EHR search program that identified all ED visits in which a patient screened positive for DV. What screening tool was used? Was the answer to only 1 DV question (out of however many DV questions were asked) considered a positive screen?

Only one screening question was used at this hospital (“are you being physically hurt or threatened by someone close to you or in your living situation?”). This question has been added to the text.

3. How do you define chief complaint (I recommend calling it a chief concern; providers may feel that patients are complaining, but patients express important concerns about their health) related to DV? Does the patient have to say “I am a victim of DV” to be considered having a chief concern related to DV?

A chief concern (this change has been incorporated throughout the manuscript) related to DV was considered to be any health concern that could reasonably be inferred to have been a direct result of DV (e.g., broken arm in fight with a boyfriend).

4. Do ED patients with repeated visits for DV receive multiple DVRR?

They may, per the provider’s discretion (e.g., if the second visit is two years later with a different partner, they will more likely receive DVRR than if the second visit a follow-up the next day). Available data did not allow us to distinguish meaningful criteria for these differing types of concerns so this was not measured. This has been noted in the text.

5. Were visits by survivors of sexual assault by a partner considered to have experienced sexual assault or DV?

Both; that’s why they were noted in data collection. This has been reworded for clarity in the Methods.

6. How is gender measured? Does it include non-binary identifying patients?

Binary. Only one DV-related record was identified as having a gender non-binary patient – and out of thousands of survivors that itself is interesting– but that obviously wasn’t sufficient for analysis. This has been noted in the text.

7. How is race defined? By patient or provider? What about a patient who has more than 1 race?

By patient, and racial and ethnic information were recorded separately. As a result, the hospital records on race and ethnicity offered very granular information. However, for analysis, only Black and white patient groups had sufficient sample sizes for analysis, as well as survivors of any race who identified their ethnicity as “Hispanic/Latinx.” When an included racial and ethnic label were recorded for a survivor, the ethnic label was given preference (e.g., a survivor who identified as “race: white, ethnicity: Latinx” was considered “Latinx” for this study). Survivors who identified with any racial or ethnic category not included in “white” “Black” or “Latinx” groups, including multiracial survivors, were not included for analysis.

8. When you describe “the relationship between perpetrator and victim”, aren’t all DV cases relationships between partners? Is this referring to sexual assault cases?

Added: For the hospital DV screening question and in this study, DV included violence perpetrated by current or former intimate partners, first-degree family members (e.g., siblings, grandparents) and roommates.

9. Do you have a measure of how well DVRR was administered? Like full 20 question Danger Assessment, or 5-item Danger Assessment screen described above? Was there safety planning conducted in the ED for positive Danger Assessment screens?

Added/edited: it also includes the 20-question Danger Assessment, a validated questionnaire that predicts a survivor’s risk of being killed by their intimate partner (52). Answers to all Danger Assessment and other DVRR questions are required prior to form submission. 

10. Were there duplicates for survivor full names and hospital visit dates at DV agency?

Not to our knowledge, though this is possible and noted in the limitations.

Figure 1: How do you define known DV? What are your inclusion criteria? Do we know why some patients did not receive DVRR? Shouldn’t they all have?

Added: These survivors were considered to have “Known DV;” this included violence perpetrated by current or former intimate partners, first-degree family members (e.g., siblings, grandparents) and roommates.

Added: Within this hospital, DVRR was available to providers at their discretion as an alternative to faxed, paper-based mandatory reports.

Data analysis

1. Does DVRR administration mean that DVRR was documented to have occurred during the ED visit? Or that a DVRR report existed in the online system? Did patients receive multiple DVRR over multiple visits?

Added: We documented DVRR administration based on a completed report in the online system; we also considered that survivors with a partial, incomplete DVRR record had not received DVRR.

Some repeat survivors received DVRR more than once, as described in the following clarification: [Repeat] survivors may have required separate intervention for distinct DV episodes, or their visit may have been a follow-up or continuation of an earlier visit and not required additional DV intervention; because this distinction could not be determined for repeat survivors, we document the number of visits by patients who qualified for study inclusion more than once.

2. Describe more about what predicted probably adds to the logistic regression models

This paragraph has been expanded as requested.

Characteristics of the study population

1. 74.2% had no other visits to the trauma center during the study period. Does that mean no other ED visits? Or no other visits for DV?

DV-related visits. This has been clarified in the text.

2. You write that survivors who experienced sexual assault were less likely to have administered DVRR. Do those patients receive a different intervention?

Yes. From line 175: We also noted the number of DV-related visits that included sexual assault, as these individuals received a separate intervention protocol that included referrals to services and advocacy intervention Added: independent of DVRR.

3. I’d recommend moving this paragraph to results.

The paragraph has been moved.

Table 1: what are the **denoting? A level of p value? Or just a statistically significant difference in % values in that characteristic?

A level of p-value (*p0.05; **p0.01). This was a typographical error and has been corrected in the text.

Results

1. Can you explain how the interaction terms increased the odds of connecting to advocacy services?

Revised text (comparable for both interaction terms): A significant interaction (DVRR x Black) indicated that DVRR was associated with a greater change for Black survivors than for white or Latinx survivors; Black survivors who received the intervention experienced an additional 4.66-fold (95% CI: 3.09-7.04) increase…

2. You write odds of accessing advocacy services, and odds of connecting to advocacy services. Are those the same things?

Yes. The language has been standardized.

3. Explain what the predicted probability adds to the odds ratio analyses already presented. Isn’t it showing similar results to DVRR associated with increased odds of receiving advocacy services?

Yes. Predicted probability more effectively illustrates the change in between-group differences with/without DVRR. This also allows the multidimensional relationship to be graphically depicted clearly. 

Table 2: is this a multinomial logistic regression? Of a series of logistic regressions shown in different columns? What is the reference group for Baseline odds, No DVRR white female? Why in the DVRR female white and latinx are white and latinx grouped together?

The headings in this table have been adjusted to address these concerns.

Discussion:

1. 2nd paragraph: You write that DVRR is an automatic referral. Given that only 54% of patients received it (from Methods/Characteristics of the study population), and your later description of providers choosing to administer DVRR may be more likely to connect survivors to services, I think you can describe this as a digital referral with lethality assessment.

The text has been amended.

2. Is there literature on Danger Assessment being validated in men who experienced violence?

Not to my knowledge; this has been noted in the text.

Strengths and limitations

1. You write that “it confidentially links the hospital and advocacy records of DV survivors.” Do you mean DVRR, or this research study on DVRR?

Revised text: This study is further unique within DV research as its methods confidentially link hospital and advocacy records of DV survivors.

2. Do you have data on how many identical given names (duplicates) occurred in your data? That would allow you to determine if duplicates affected intervention and non-intervention groups equally. I otherwise do not think you can conclude this.

These data were not collected; the conclusion has been deleted. 

Reviewer #2: Access to domestic violence advocacy by race, ethnicity and gender:

The impact of a digital warm handoff from the emergency department

Accept minor revisions

This study fills an important gap in our evidence base regarding intervention for IPV, and its focus on racial differences is particularly timely. It is also very nice to see an intervention that differentially and positively impacts Latinx and Black women. The intervention itself is well-conceived to address challenges in IPV referral in the ED, and in particular that the hand-off is to an advocacy organization rather than only to law enforcement (as required by CA law).

Specific comments:

1. The introduction is far too lengthy and reads more like an essay. Only the salient points for this analysis, summarized, are needed.

The introduction has been substantially shortened.

2. The authors state in their methods that they detected no missing or incomplete data, but with only aggregate data from the advocacy organizations, the quality of the data is impossible to verify. This should be noted.

Added: We detected no missing or incomplete data within the hospital dataset; due to its aggregate nature the quality of the advocacy data is impossible to verify.

3. The process for linking referrals to follow-up visits is not described. Linkages based on name, because of spelling variability, are particularly problematic and generally have some validity challenges.

This is true, and is discussed in the manuscript. Added: Because records were matched based on survivor first and last names, spelling variability between hospital and advocacy records may also compromise the validity of the matched records.

4. Given strong evidence that BIPOC patients report challenges in their interactions with healthcare facilities, the generalizability of this study needs to be addressed. For example, many EDs might not have a majority of Black IPV patients (in this sample, 63.5%), and perhaps the staff at this hospital are particularly adept in culturally appropriate and ethnicity-centered healthcare. How would this potentially impact EDs with different patient profiles?

Added: In addition, the hospital in this study saw a majority of Black IPV patients (63.5%); this proportion is likely higher than in many other hospitals and may be associated with a higher level of culturally appropriate healthcare. The impact of DVRR for racial and ethnic minority patients in hospitals with a less heterogenous patient population may differ from the findings of those in this study due to potential differences in care.

5. Although these findings are very promising, nearly half of the IPV victims in the ED did not receive the intervention, and even with the improved access to advocacy services with the intervention, only between 23% and 43% of those referred accessed services. The discussion should address these issues by providing some insight into why these proportions are low and how they might be improved.

Added: Reasons for this may range from non-working phone numbers to unanswered phone calls. These missed connections may have resulted from survivors changing their mind about receiving advocacy care, the abuser monitoring the survivor’s phone, or inadvertent typos or missed calls.

---

## [Decision Letter · Decision Letter 1]

28 Oct 2021

PONE-D-21-05014R1Access to domestic violence advocacy by race, ethnicity and gender:

The impact of a digital warm handoff from the emergency departmentPLOS ONE

Dear Dr. Brignone,

Thank you for submitting your manuscript to PLOS ONE. After careful consideration, we feel that it has merit but does not fully meet PLOS ONE’s publication criteria as it currently stands. Therefore, we invite you to submit a revised version of the manuscript that addresses the points raised during the review process.

We look forward to receiving your revised manuscript.

Kind regards,

Michelle L. Munro-Kramer, PhD, CNM, FNP-BC

Academic Editor

PLOS ONE

Journal Requirements:

Additional Editor Comments (if provided):

Thank you for your careful attention to the previous comments and suggestions. There are a few very minor remaining comments from Reviewer #1. I look forward to accepting this manuscript after these have been addressed.

Reviewers' comments:

Reviewer's Responses to Questions

**Comments to the Author**

1. If the authors have adequately addressed your comments raised in a previous round of review and you feel that this manuscript is now acceptable for publication, you may indicate that here to bypass the “Comments to the Author” section, enter your conflict of interest statement in the “Confidential to Editor” section, and submit your "Accept" recommendation.

Reviewer #1: (No Response)

Reviewer #2: All comments have been addressed

2. Is the manuscript technically sound, and do the data support the conclusions?

Reviewer #1: Yes

Reviewer #2: Yes

3. Has the statistical analysis been performed appropriately and rigorously? 

Reviewer #1: Yes

Reviewer #2: Yes

4. Have the authors made all data underlying the findings in their manuscript fully available?

Reviewer #1: Yes

Reviewer #2: Yes

5. Is the manuscript presented in an intelligible fashion and written in standard English?

Reviewer #1: Yes

Reviewer #2: Yes

6. Review Comments to the Author

Reviewer #1: This is an important study of ED patients with DV, their receipt of a digital warm handoff intervention called DVRR, and their receipt of advocacy services. I have several questions and comments to improve clarity.

Abstract:

1. Background: I suggest changing first sentence to “racial/ethnic minority survivors of domestic violence (DV) referred from emergency departments (Eds) face barriers connecting with advocacy services due to systemic inequalities.” Their inequities in healthcare access may not be the same as those found in advocacy services. I also suggest describing DVRR as an electronic reporting system to fulfill mandatory reporting requirements to law enforcement, with report also going to DV advocacy agency.

2. Methods: Describe time frame (2014-2018) for study. Briefly describe how DV-associated ED visits were defined. I recommend not describing sample sizes (1366, 740, 323) in methods section as those are results. You could state something like “We assessed 2014-2018 ED visit chief concern for DV, measured if DVRR occurred, and if patient names matched DV advocacy agency records of survivors using services”. I would suggest giving sample size for number used in logistic regression analyses and number used in predicted probability analyses.

Manuscript text

Introduction

1. 2nd paragraph: You write that “standard of care for DV often does not extend beyond hospital screening; when it does, it typically consists of printed educational material or a phone number to a community-based advocacy agency…”, and I think you are meaning that standard of care for DV includes identification of DV but does not necessarily include providing support and resources, or follow-up to see if patient used advocacy agency services. I would clarify this.

2. 2nd paragraph: You write “…warm handoff, in which the ED provider personally transfers the survivor’s DV care to a DV advocate.” A transfer of care in medical care more typically means physical movement of the patient location. A warm handoff might be better described as a healthcare provider describing DV advocacy agency services and possibly calling the DV advocacy agency while in the patient room. This may include a DV advocacy agency staff member coming to the patient location for an in-person provision of support and resources. I would clarify this.

3. Race and ethnicity, and Gender sections: I recommend condensing this into 1 paragraph. The focus of the paper is ED visits, DVRR and DV advocacy visit. The first 2 paragraphs describe ED visits and DV advocacy, and the last paragraph describes DVRR.

Results

1. I recommend changing “Analysis” header (between first and second paragraphs) to “DVRR and receiving advocacy services by race/ethnicity and gender”. Otherwise, the reader will think this is the methods section analysis plan

2. Last paragraph: what does likelihood difference assist reader in knowing?

Table 2:

1. I would put 95%CI with odds ratio in 1 row.

2. Why is the reference group baseline odds ratio (white, female) 0.12? Shouldn’t it be 1.0 as it’s the reference group?

Figures 1 and 2: what does statistical test for % increase add? I think you can remove it. It’s otherwise too much information. It’s obvious from the table that there is a visually large increase by DVRR.

S1 Appendix: can you explain more why this appendix is needed? Doesn’t Figure 2 adequately show differences in DVRR status by race and ethnicity?

Discussion: Paragraph 3: the authors describe how 5-question Danger Assessment “is an important step toward improving the strength of the Danger Assessment component of DVRR in reducing bias and providing culturally appropriate care.” I think the using 5-question DA will help provider time in completing the DA, and therefore completing the full DVRR assessment. It may increase DVRR administration from 54% to something higher.

Strengths and limitations: Paragraph 3: can you describe the “data restrictions that made it impossible to study additional groups that…experience heightened barriers to DV response or services”. Can you explain more about those data restrictions?

Reviewer #2: The revision addresses my original comments, and I have no further comments.

The revision addresses my original comments, and I have no further comments.

7. PLOS authors have the option to publish the peer review history of their article (what does this mean?). If published, this will include your full peer review and any attached files.

Reviewer #1: No

Reviewer #2: No

---

## [Author Response · Author response to Decision Letter 1]

10 Dec 2021

The authors would like to thank the reviewers/editor for their thoughtful, constructive comments. Their suggestions have been implemented as described below.

Reviewer #1: This is an important study of ED patients with DV, their receipt of a digital warm handoff intervention called DVRR, and their receipt of advocacy services. I have several questions and comments to improve clarity.

Abstract:

1. Background: I suggest changing first sentence to “racial/ethnic minority survivors of domestic violence (DV) referred from emergency departments (Eds) face barriers connecting with advocacy services due to systemic inequalities.” Their inequities in healthcare access may not be the same as those found in advocacy services. I also suggest describing DVRR as an electronic reporting system to fulfill mandatory reporting requirements to law enforcement, with report also going to DV advocacy agency.

The proposed changes have been implemented.

2. Methods: Describe time frame (2014-2018) for study. Briefly describe how DV-associated ED visits were defined. I recommend not describing sample sizes (1366, 740, 323) in methods section as those are results. You could state something like “We assessed 2014-2018 ED visit chief concern for DV, measured if DVRR occurred, and if patient names matched DV advocacy agency records of survivors using services”. I would suggest giving sample size for number used in logistic regression analyses and number used in predicted probability analyses.

Each of these proposed changes has been implemented.

Manuscript text

Introduction

1. 2nd paragraph: You write that “standard of care for DV often does not extend beyond hospital screening; when it does, it typically consists of printed educational material or a phone number to a community-based advocacy agency…”, and I think you are meaning that standard of care for DV includes identification of DV but does not necessarily include providing support and resources, or follow-up to see if patient used advocacy agency services. I would clarify this.

The paragraph has been clarified per this exceptionally helpful comment.

2. 2nd paragraph: You write “…warm handoff, in which the ED provider personally transfers the survivor’s DV care to a DV advocate.” A transfer of care in medical care more typically means physical movement of the patient location. A warm handoff might be better described as a healthcare provider describing DV advocacy agency services and possibly calling the DV advocacy agency while in the patient room. This may include a DV advocacy agency staff member coming to the patient location for an in-person provision of support and resources. I would clarify this.

The paragraph has been revised to clarify this information.

3. Race and ethnicity, and Gender sections: I recommend condensing this into 1 paragraph. The focus of the paper is ED visits, DVRR and DV advocacy visit. The first 2 paragraphs describe ED visits and DV advocacy, and the last paragraph describes DVRR.

These two sections have been condensed into a single section with three paragraphs. The key findings in this paper highlight differential results by race, ethnicity and gender and this section has been condensed to more concisely presage those findings.

Results

1. I recommend changing “Analysis” header (between first and second paragraphs) to “DVRR and receiving advocacy services by race/ethnicity and gender”. Otherwise, the reader will think this is the methods section analysis plan

The heading has been revised per this section.

2. Last paragraph: what does likelihood difference assist reader in knowing?

This paragraph has been revised to read “Female survivors were significantly more likely than male survivors to subsequently connect with advocacy services (Fig 3). DVRR use appears to narrow that gap within this sample.”

Table 2:

1. I would put 95%CI with odds ratio in 1 row.

This table has been reformatted and the preceding paragraph has been updated accordingly

2. Why is the reference group baseline odds ratio (white, female) 0.12? Shouldn’t it be 1.0 as it’s the reference group?

This information has been removed from the table and inserted in the preceding paragraph

Figures 1 and 2: what does statistical test for % increase add? I think you can remove it. It’s otherwise too much information. It’s obvious from the table that there is a visually large increase by DVRR.

These tests have been removed.

S1 Appendix: can you explain more why this appendix is needed? Doesn’t Figure 2 adequately show differences in DVRR status by race and ethnicity?

This appendix has been removed.

Discussion: Paragraph 3: the authors describe how 5-question Danger Assessment “is an important step toward improving the strength of the Danger Assessment component of DVRR in reducing bias and providing culturally appropriate care.” I think the using 5-question DA will help provider time in completing the DA, and therefore completing the full DVRR assessment. It may increase DVRR administration from 54% to something higher.

This consideration has been added. 

Strengths and limitations: Paragraph 3: can you describe the “data restrictions that made it impossible to study additional groups that…experience heightened barriers to DV response or services”. Can you explain more about those data restrictions?

The sentence described has been revised and expanded to state the following:

To protect client privacy at the advocacy agency, covariate data collection was restricted a priori to variable combinations with cell sizes of 2 or greater. As a result, researchers were limited to six major covariate indicators (DVRR administration, male or female gender, Black, Latinx or non-Latinx white race/ethnicity). This required excluding certain populations with small sample sizes (e.g., Asian, Pacific Islander, Indigenous), and not specifying other relevant characteristics either due to a small sample size (e.g., LGBTQ+ identity) or non-categorical data structure (e.g., age).

---

## [Editor Report · Decision Letter 2]

18 Feb 2022

Access to domestic violence advocacy by race, ethnicity and gender:

The impact of a digital warm handoff from the emergency department

PONE-D-21-05014R2

Dear Dr. Brignone,

We’re pleased to inform you that your manuscript has been judged scientifically suitable for publication and will be formally accepted for publication once it meets all outstanding technical requirements.

Kind regards,

Michelle L. Munro-Kramer, PhD, CNM, FNP-BC

Academic Editor

PLOS ONE

Additional Editor Comments (optional):

Thank you for the revisions and congratulations on the acceptance of this article. It will make a significant contribution to the literature.
---

## [Editor Report · Acceptance letter]

10 Mar 2022

PONE-D-21-05014R2 

Access to domestic violence advocacy by race, ethnicity and gender:
The impact of a digital warm handoff from the emergency department 

Dear Dr. Brignone:

I'm pleased to inform you that your manuscript has been deemed suitable for publication in PLOS ONE. Congratulations! Your manuscript is now with our production department. 

Kind regards, 

on behalf of

Dr. Michelle L. Munro-Kramer 

Academic Editor

PLOS ONE